# Did you donate? Talking about donations predicts compliance with solicitations for donations

Joris Melchior Schröder[1,2]*, Eva-Maria Merz[1,2], Bianca Suanet[1],
Pamala Wiepking[1,3]

**1** Department of Sociology, Vrije Universiteit Amsterdam, Amsterdam, The Netherlands, **2** Department of Donor Medicine Research, Sanquin Research, Amsterdam, The Netherlands, **3** Lilly Family School of Philanthropy, Indiana University, Indianapolis, IN, United States of America

☯ These authors contributed equally to this work.
* j.m.schroeder@vu.nl

## Abstract

Many forms of prosocial behaviour are highly institutionalized. They are facilitated by organizations that broker between donors and recipients. A highly effective tool that organizations use to elicit prosocial behaviour are solicitations for donations (e.g., of blood, time, or money). Using register and survey data on blood donations in the Netherlands, we examine to what extent compliance with these solicitations is predicted by being recruited via word of mouth (WOM) and talking about donations. Our model predicts that donors that are one unit higher on our measure of talking about donations (range = 1–4) have a 2.9 percentage points higher compliance with solicitations for donations. In addition, this association is stronger for novice donors. Our study demonstrates the social embedding of the donors' decision-making processes about compliance. For practice, our results imply that organizations may increase their contributors' communication about donations to increase the effectiveness of their solicitations.

**Data Availability Statement:** Data cannot be shared publicly because it contains personal identifiers and sensitive information on health. Data are available from Sanquin Research (contact via https://www.sanquin.org/research/donor-insight/

## 1. Introduction

Many forms of prosocial behaviour are highly institutionalised. They are structured and facilitated by organizations that serve as brokers between donors and recipients [1, 2]. One of the key tools that organizations use to elicit prosocial behaviour are solicitations for donations [3–5]. These solicitations are a powerful factor determining an individual's prosocial behaviour in the forms of blood donations [4, 6], charitable giving [7–9], and volunteering [5, 10]. Nevertheless, not every solicitation results in a donation, which raises the question of what determines their effectiveness.

Research has shown that an individual's probability of compliance with a solicitation depends on the content and the procedure of solicitations themselves [11–14]. However, even when solicitations are uniform in procedure, format, and content, there are large variations in the compliance with solicitations between individuals and social contexts. There is some evidence that these variations are explained by the characteristics of individuals (e.g., their socio-demographic characteristics, perceptions, and attitudes) [6, 10], and by characteristics of the

collaboration/proposal-for-analyses or email: donorinsight@sanquin.nl) for researchers who meet the criteria for access to confidential data.

**Funding:** This work was supported by the European Research Council (ERC) under the European Union's Horizon 2020 research and innovation programme [grant agreement No. 802227 to E.-M.M.] (funder website: https://erc.europa.eu/). Pamala Wiepking's position at the Lilly Family School of Philanthropy is funded by the Stead Family (no website), her work at the Vrije Universiteit Amsterdam is funded by the Dutch Postcode Lotteries (funder website: https://www.novamedia.com/). The funders had no role in study design, data collection and analysis, decision to publish, or preparation of the manuscript.

**Competing interests:** The authors have declared that no competing interests exist.

social and physical context (e.g., presence of opportunities to give or social norms related to prosocial behaviour) [4]. Previous research further shows that social contexts that allow for social influences to be at work promote prosocial behaviour [15]. For example, simply allowing for communication between (potential) contributors is a highly effective strategy for increasing contributions in public goods games [16]. In addition, individual exposure to social influences has been shown to affect blood donation behaviour [17, 18]. These studies highlight that the decision about prosocial behaviour is dependent on the link between individuals and their social environment. But even though most donations are made in response to a solicitation, we know little about how this social embeddedness shapes compliance behaviour.

In this study, we analyse the social embeddedness of compliance behaviour by looking at two factors linking donors to their social environment: talking about donations and being recruited via word of mouth (WOM). To conceptualize donor behaviour and to acknowledge that factors at different levels (e.g., the social and physical context, and individual characteristics) impact the individuals' decision to donate, we develop a theoretical model that integrates existing theories of donor decision-making into social-ecological systems (SES) analysis [19–21]. Empirically, we study compliance with solicitations to donate whole blood in the Netherlands. We make use of register data on about 157 000 solicitations for donations and a large-scale survey among a sample of 24 045 registered blood donors in the Netherlands, which was linked on the individual level.

The main contribution of this paper is twofold: First, we recognize solicitations for donations as a distinct level of analysis and demonstrate the social embedding of the decision-making process about compliance with these solicitations. Higher talking about blood donations is associated with higher compliance with solicitations for donations, and especially so among novice donors. This is in line with previous findings on social influences on general prosocial behaviour, and especially studies that have highlighted the importance of communication for increasing contributions to public goods. The interaction effect with experience further shows the importance of habit formation in the decision-making process, as external (social) influences seem to become less relevant as compliance becomes increasingly habitual. In addition, our study illustrates the differential role of altruistic values for compliance rather than general prosocial behaviour. For compliance behaviour, social influences in the form of talking about donations do not seem to be more important for the compliance of individuals with low altruistic values. This is in contrast to empirical findings on general prosocial behaviour, where social influences are typically more important for those with lower altruistic values [22, 23]. In addition, exploratory analysis shows that altruistic values are overall negatively associated with compliance.

Second, our study informs the practice of organizations dependent on the effectiveness of solicitations for donations. Blood has a limited shelf life and demand is changing continuously. Knowing about the factors that determine the compliance with solicitations for donations is therefore essential to determine the number of necessary solicitations to ensure a sufficient blood stock. Our results imply that organizations could invest in communication about donations and/or recruitment among potentially more communicative donors to increase the compliance with solicitations for donations and increase their effectiveness. One such interventions could be group-donation programmes, where donors can join a group of donors that they can talk to and donate with.

## 2. A social-ecological systems model of compliance with solicitations for donations

To conceptualize compliance behaviour, we draw on a SES framework [19–21, 24]. A core characteristic of SES analysis is the observation that individuals' decision-making is influenced

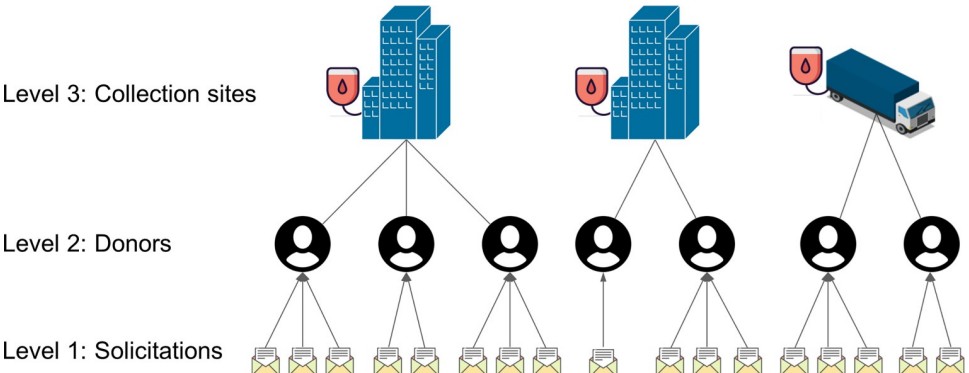

**Fig 1. Levels of analysis: solicitations, individuals, and collection sites.** Buildings represent fixed collection sites. The truck represents a mobile collection site.

by the social and physical context that they are embedded in [20, 21]. In the case of blood donations, the collection sites are one social context relevant to the decision-making process about compliance behaviour. Blood donations require physical presence at a collection site, and these sites are therefore spaces where different donors experience the same facilities, the same staff, and where donors might meet and communicate with each other. In the Netherlands, all blood donations are collected by the non-profit organization Sanquin and are voluntary and non-remunerated. Prospective donors register to become a donor with the blood bank, for example after being recruited by a friend. After registration, donors undergo an initial health screening and, if they are eligible, are added to the donor database. Subsequently, donors are repeatedly solicited to make a donation at a specific collection site (typically close to where they live or work), based on the current demand for their blood type. Fig 1 illustrates this structure: solicitations to donate are sent out to donors, who make donations at a specific collection site.

At the same time, the SES perspective views individuals as autonomous agents. Their decisions are based on individual characteristics such as perceptions, socio-demographics, and attitudes. We build on the SES framework by Schlüter et al. [21], where decision-making is captured by the four processes of perception, evaluation, selection of behaviour, and behaviour. These components are central to multiple theories of human decision-making, and they form the basis for our theoretical model of compliance behaviour within the social and physical context of the collection sites, which is depicted in Fig 2. The dashed outer box represents the collection sites as the social and physical context, with its characteristics in the blue inner box. The individual is represented by the dashed inner box, with their characteristics given in the red inner box. In this model, decision-making (represented by ellipses) starts with an individual's perception of their social and physical environment. The individual then evaluates new information and potentially updates their own characteristics based on these inputs. The individual's characteristics feed into the process of selecting a behaviour. Finally, a behaviour is executed and potentially affects the characteristics of the social and physical environment. Below, we develop a model of compliance behaviour and state hypotheses about its social embeddedness, which we will test in the empirical analysis.

## 2.1. Perception

Perception is the process by which an individual senses the surrounding social and physical environment [21]. Important components of perception of the social environment are social

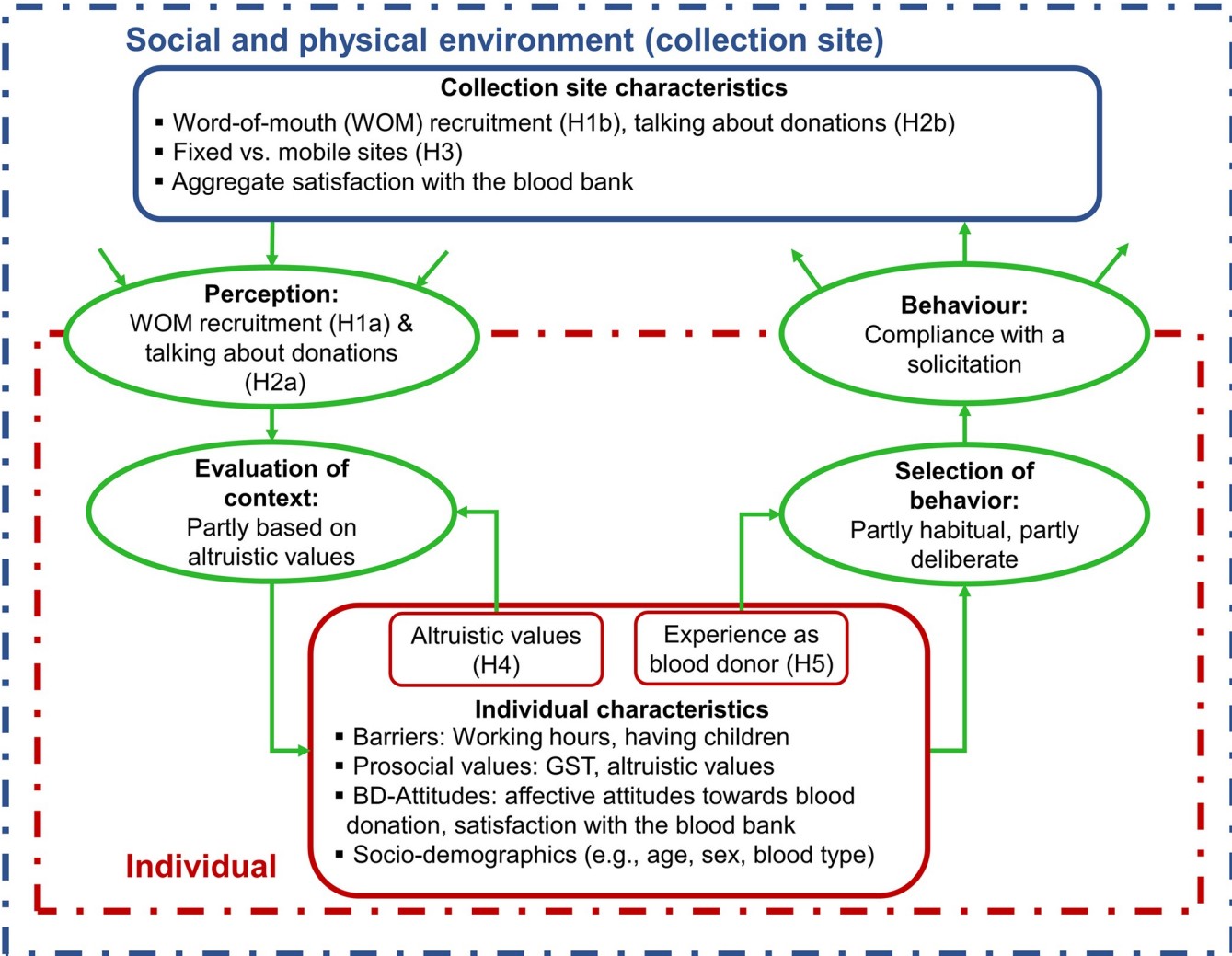

**Fig 2. Conceptual model of compliance with solicitations for donations.** Structural elements are depicted as boxes, and elements of the decision-making process are depicted as ellipses.

interactions and communication. For example, norms and expectations for behaviour emerge via communication and interaction among individuals [15, 25]. We consider two factors of perception that link individuals to their social environment, namely being recruited via WOM, and talking about donations.

### Recruitment and communication–individual level

Becoming a blood donor typically starts with being recruited by either the blood bank or via WOM, that is, informal person-to-person communication [26, 27]. If one is recruited via WOM, the recruiters are typically partners, family or friends. In comparison to donors recruited via other channels, donors that were recruited via WOM might additionally be motivated by social influences from their recruiters and therefore be more likely to comply with a solicitation. In line with this argument, Piersma & Klinkenberg [28] have shown that blood donors recruited by other donors have a higher donation frequency than those that signed up on their own initiative or because of promotions by the blood bank. In comparison to their

study, however, WOM recruitment captures a broader range of recruitment through others, because these others are not necessarily donors themselves. Nevertheless, we expect a positive association and hypothesise:

*Hypothesis 1a: The probability of compliance with a solicitation for a donation is higher for donors that were recruited via word of mouth.*

Communication among individuals is one of the most successful strategies for promoting cooperation and prosocial behaviour [16, 24]. Explanations for the effectiveness of communication include that it conveys information about need, expectations about others' behaviour, norms and their enforcement, and that it fosters group identities and emotions which promote donations [15, 16, 29, 30]. In the case of blood donations, talking about donations might convey such motivations. The more a donor talks about donations, the more they should be affected by such motivations. As such, we hypothesise:

*Hypothesis 2a: The probability of compliance with a solicitation for a donation is higher for donors that talk more about blood donations.*

## Recruitment and communication–collection site level

In addition to variation on the individual level, the level of WOM recruitment and talking about donations might vary between collection sites. These differences might affect compliance rates irrespective of whether a donor was recruited via WOM or talks about donations. For example, talking about blood donations can increase the salience of a social norm for blood donation. In addition, social interactions related to blood donations provide an opportunity for indirect reciprocity to occur, where a person who helps another later receives help from a third person (not necessarily in the same domain) [31, 32]. A higher proportion of people recruited via WOM, as well as more talking about blood donations on the collection-site level might indicate that blood donorship is more closely tied to these social mechanisms, even if an individual donor might not be embedded in a network of blood donors themself. The proportion of donors recruited via WOM, and the average talking about donations at the collection site level should thus be positively related to the individual's compliance with a solicitation.

*Hypothesis 1b: The probability of compliance with a solicitation for a donation increases with the proportion of donors recruited via word of mouth at the collection site level.*

*Hypothesis 2b: The probability of compliance with a solicitation for a donation increases with the average talking about donations at the collection-site level.*

## The role of fixed versus mobile collection sites

Social proximity has been shown to moderate the effect of social influences [33], including for the case of blood donations [18]. While we have no specific information on who interacts with whom, we can use information about collection sites as a proxy for social closeness among (potentially) interacting donors. In the Netherlands, blood is collected at fixed and mobile sites. Fixed collection sites are placed in larger cities and have extended opening hours. Donors registered at fixed sites are invited to donate during a two-week walk-in period starting shortly after receiving the solicitation letter. Mobile collection sites are used to collect blood in less densely populated areas such as smaller towns and villages, and therefore draw on a smaller pool of donors than fixed collection sites. Because social networks are spatially clustered [34], these donors are more likely to know other donors donating at the same collection site. In

addition, donors at mobile collection sites are invited to donate at a specific date rather than within a two-week walk-in period. Together, these factors imply that donors invited to a mobile site are more likely to meet and talk to other donors that they know. The relation between talking about donations and compliance might therefore be stronger at mobile rather than fixed donation sites.

*Hypothesis 3: The relationship between the compliance and talking about donations is moderated by the type of collection site (fixed or mobile) such that it is stronger at mobile collection sites.*

## 2.2. Evaluation

Evaluation is the "process by which an individual determines the significance, worth, or condition of the perceived state of the social and bio-physical environment" [21].

### The role of altruistic values

Simpson and Willer [15] argue that social influences are stronger on more egoistically as compared to altruistically motivated individuals. This is because more egoistically motivated individuals are typically unlikely to contribute to public goods but might be motivated by social influence. Altruistically motivated individuals, on the other hand, are more likely to contribute in the first place. In an empirical study, Simpson and Willer [23] found that egoistically motivated individuals indeed did respond more strongly to reputational incentives than altruistically motivated individuals. Similarly, Feinberg et al. [22] have shown that the threat of gossip via communication more strongly promotes cooperation of egoistically motivated than altruistically motivated individuals. If talking about donations (partly) captures gossip and reputational mechanisms, it should be primarily related to the compliance of donors with lower altruistic values.

*Hypothesis 4: The relationship between the compliance with a solicitation for a donation and talking about donations is moderated by the altruistic values of individuals, such that it is stronger for those with lower altruistic values.*

## 2.3. Selection of behaviour

Selection of behaviour is the process by which individuals choose a behaviour based on their individual characteristics [21].

### The role of experience as a blood donor

The literature on blood donation behaviour has revealed that the selection of a behavioural option becomes partly habitual over the course of a blood donor career [6, 35]. A behaviour becomes more habitual the more it is performed, which is why the number of previous donations is often used as an indicator for habit formation. Once a strong habit is developed, the decision whether or not to donate is made with little conscious deliberation [36]. External factors, such as social influence, should therefore become less relevant with more experience as a blood donor [36–38]. We hypothesise that:

*Hypothesis 5: The relationship between the compliance with a solicitation for a donation and talking about donations is moderated by experience, such that it is weaker for more experienced donors.*

## 3. Data

Our empirical analysis makes use of register and survey data. Data from the blood bank information system [39] provides information on solicitations and donations of individuals. Importantly, the use of register data mitigates problems of potential observability bias or recall bias. This data is linked to the second wave of the Donor InSight survey (DIS-II; 2012–2013, N = 34 826, for details see Timmer et al. [40]), which provides extensive information on donors' socio-demographic characteristics (e.g., age, gender) as well as their potential motivations for compliance (e.g., talking about donations, altruistic values). The unit of observation are all solicitations in 2012 and 2013 that were sent out to donors that participated in the DIS-II survey.

Our analysis involves three levels (see Fig 1): the level of solicitations, the level of individual donors, and the level of the collection sites. Donors almost always donate at the same collection sites (close to where they live or work). Hence, we assign donors to the collection site that they received the most solicitations for in 2012 and 2013 to obtain a clear hierarchical data structure. Below, we provide a list of the measured variables and the latent constructs (and scales used to measure these) that are used in our analysis. Table 1 provides descriptive statistics for the study measures.

### 3.1. Dependent variable

Compliance with a solicitation for a blood donation is a binary variable derived from the blood bank information system. For each solicitation, we track a donor's compliance four weeks after they have received a solicitation. Thus, the variable takes the value 1 if a donation was made within four weeks after receiving a solicitation, and the value 0 otherwise. In this way, we capture compliance even if a donor comes to donate slightly before or after the period or date stated on the solicitation letter. For several reasons that are unrelated to the intention to donate, such as low haemoglobin levels, blood donations might be unsuccessful. Thus, donation attempts capture individuals' prosocial behaviour better than only successful donations.

The data shows that about 57% of solicitations were followed by a donation attempt (see Table 1). This reveals lots of potential for improving compliance rates. The data further shows that there is large variation in compliance rates on the individual level (Fig 3A), and the collection site level (Fig 3B). The compliance rate at mobile collection sites (61%) is slightly higher in comparison to the compliance rate at fixed collection sites (56%).

### 3.2. Predictors of interest

*Word of mouth (WOM) recruitment* is a binary variable indicating whether donors report that they were initially recruited via WOM. The recruitment channel was elicited with the question "What made you decide to become a donor?", and the response options were "(1) own idea, (2) brochure from the blood bank, (3) recruitment activities of the blood bank, (4) newspaper, (5) internet, (6) partner, (7) family, (8) friends or acquaintances, and (9) other", where multiple response options could be selected. WOM recruitment takes the value 1 if the respondent selected at least one of the options 6, 7, or 8, and the value 0 otherwise. Since multiple response options could be selected in the survey, the recruitment channels are not mutually exclusive. Our main analysis uses the variable described above indicating whether a donor was recruited via WOM, potentially among other influences on recruitment. As a robustness check, we also constructed a variable indicating whether a donor was *only* recruited via WOM. The results are reported in S7 Table, and do not substantially differ from those in our main analysis.

**Table 1. Descriptive statistics of study measures.**

| Variable | Mean | SD | Min. | Max. | N |
|---|---|---|---|---|---|
| *Solicitation level* | | | | | |
| Compliance | 0.572 | 0.495 | 0 | 1 | 157017 |
| *Individual level* | | | | | |
| WOM recruitment | 0.458 | 0.498 | 0 | 1 | 23862 |
| Talking about donations | 2.011 | 0.471 | 1 | 4 | 24016 |
| Working hours | 26.604 | 16.116 | 0 | 99 | 22883 |
| Having children | 0.735 | 0.442 | 0 | 1 | 24045 |
| GST | 3.413 | 0.712 | 1 | 5 | 23972 |
| Altruistic values | 3.685 | 0.693 | 1 | 5 | 23967 |
| Awareness of need | 4.635 | 0.624 | 1 | 5 | 24001 |
| Affective attitudes | 3.677 | 0.769 | 1 | 5 | 23597 |
| Satisfaction with blood bank | 4.437 | 0.553 | 1 | 5 | 24015 |
| More solicitations | 0.138 | 0.345 | 0 | 1 | 23953 |
| Less solicitations | 0.024 | 0.152 | 0 | 1 | 23953 |
| Age | 48.176 | 12.989 | 18 | 71 | 24045 |
| Male | 0.443 | 0.497 | 0 | 1 | 24045 |
| Previous donations | 32.689 | 26.736 | 1 | 256 | 24045 |
| Common blood type | 0.674 | 0.469 | 0 | 1 | 24044 |
| Rare blood type | 0.195 | 0.396 | 0 | 1 | 24044 |
| Universal blood type | 0.131 | 0.337 | 0 | 1 | 24044 |
| *Collection site level* | | | | | |
| Mobile | 0.652 | 0.478 | 0 | 1 | 164 |

Notes: Values for latent constructs (GST, altruistic values, awareness of need, affective attitudes, and satisfaction with the blood bank) are mean-scores based on the underlying items.

To measure *talking about donations*, DIS-II participants were asked: "How often do you speak with people in your circle of acquaintances about blood donation?", and response options were "never", "occasionally", "regularly", and "often", which were coded as 1–4.

Conceptually, WOM recruitment and talking about donations are not necessarily related. WOM recruitment refers to the way donors came to the initial decision about registering with

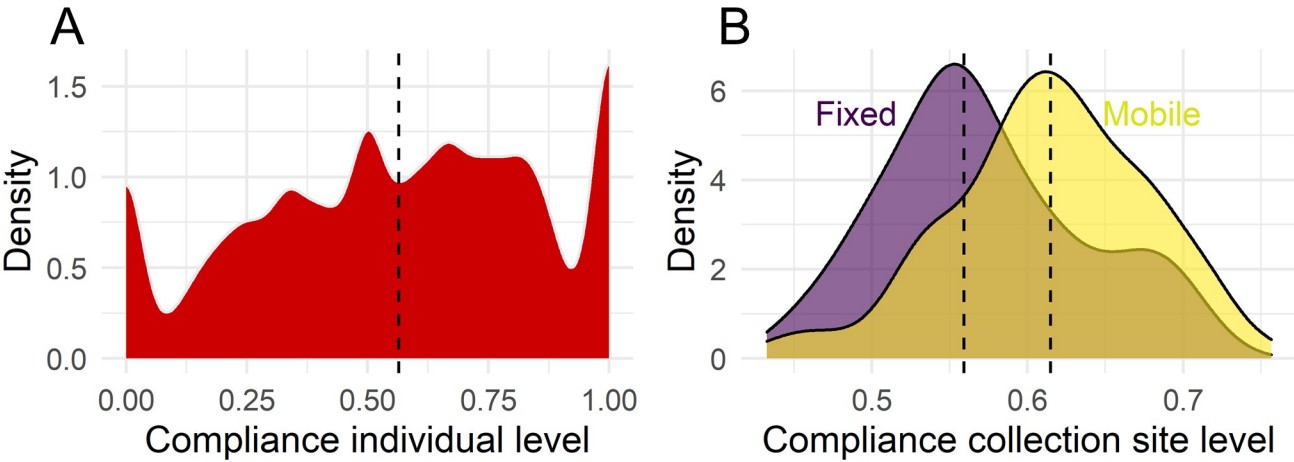

**Fig 3. Compliance with solicitations for whole blood donations in 2012 and 2013.** The dashed lines indicate the means of each distribution. (A) Compliance on the individual level, N = 24 016 individuals. (B) Compliance on the collection site level, N = 164 collection sites.

the blood bank. Talking about donations, in contrast, is about how much people speak with their entire circle of acquaintances about blood donations at the time of taking the survey. As such, the two predictors differ conceptually, but also with respect to time. For many donors, the initial decision to register as a blood donor will have taken place years ago. Having been recruited via WOM does therefore not say much about the level of talking about donations in subsequent years. This is also shown by a weak correlation between these two variables in the data (with a Pearson correlation coefficient r = 0.02, see S3 Table).

## 3.3. Moderators

*Experience as a blood donor* is measured by the overall number of donations recorded in the blood bank information system up to the date of response to the DIS-II survey.

*Altruistic values* are measured using a 5-point Likert scale based on three items. The items originate in the Survey of Interpersonal Values (SIV) by Gordon, and were translated into Dutch by Drenth & Kranendonk [cited in 41]. The items are: "I try to work towards the wellbeing of society.", "It is important to me that I help others.", and "I think it is important to help the poor and the needy.".

*Mobile* is a binary variable on the collection site level derived from Sanquin records that takes the value 1 if a collection site was mobile, and 0 otherwise.

## 3.4. Covariates

There are three groups of factors that have been shown to affect the decision-making process about compliance, and that are likely also related to WOM recruitment and to talking about donations. They are thus (potential) confounders of the relationship between WOM recruitment or talking about donations and compliance with solicitations for donations. We therefore consider them as covariates. Below, we describe the measurement of the covariates. Section 4 describes how these variables are included in the statistical models.

First, because blood donations require time-investment, there are opportunity costs of compliance [10, 42]. We include *weekly working hours* and *having children* (no = 0, yes = 1) as two indicators for these costs.

Second, several values have been shown to affect prosocial behaviour more generally, among them *awareness of need*, *generalized social trust (GST)*, and *altruistic values* [8]. There is no established scale to measure *awareness of need for blood donations*, but it has been identified as a central factor motivating charitable giving [8], and it is also among frequently self-reported motivations for donating blood [43]. We therefore use the responses to the item "My blood is needed." on a five-point scale ranging from "totally disagree" to "totally agree" to measure awareness of need for blood donations. Our measure of *generalised social trust (GST)* was originally developed by Rosenberg [44], and adapted by Bekkers [45]. GST is measured using a 5-point Likert scale based on two items: "In general, most people can be trusted.", and "You cannot be careful enough when you are dealing with other people.".

Third, blood donation specific attitudes affect compliance. *Satisfaction with the blood bank* is a central influence on blood donation intentions [46] but not self-reported compliance with invitations to donate [4]. Following Merz et al. [41], we measure satisfaction with the blood bank using the following four items measured on a five-point Likert scale ranging from "totally disagree" to "totally agree": "I think the blood bank is a professional organization.", "There is sufficient opportunity to ask questions at the blood bank.", "I am convinced that the blood bank treats my personal information with care.", "I am approached personally at the blood bank.". *Affective attitude towards blood donation* is a variable from the extended Theory of Planned Behavior (TPB) [47] that has been adapted to the blood donation context [48].

Affective attitude was elicited using a 5-point semantic-differential scale including three statements in response to the statement "I find giving blood. . .": "pleasant–unpleasant", "annoying–enjoyable", and "unappealing–appealing". A perception of receiving too few or too many solicitations likely affects donors' motivation to comply [49]. To measure the feeling of receiving too few or too many invitations, participants were asked: "Are you satisfied with the number of times per year that you receive an invitation to donate or are able to make an appointment to donate?", where response options were "yes", "no, I would like to receive an invitation/make an appointment more often", and "no, I would like to receive an invitation/make an appointment less often", and "no opinion/not applicable". We create a binary variable indicating whether a donor *wants more solicitations*, and another binary variable indicating whether a donor *wants less solicitations.*

As further sociodemographic covariates we include *age* in years, and being *male* (no = 0, yes = 1). Finally, we include a measure for the blood type, because the frequency of receiving solicitations depends on the blood type. Blood type is included in the three categories '*rare blood type*' (B+, AB+, A-, B-, AB-), '*common blood type*' (0+, A+), and '*universal blood type*' (0-).

## 4. Methods

This study was approved by the Research Ethics Review Committee (RERC) of the Faculty of Social Sciences, Vrije Universiteit Amsterdam [reference number RERC/18-10-08]. The Donor InSight study was approved by the Medical Ethical Committee Arnhem-Nijmegen in the Netherlands [CMO-nr: 2005/119]. All participants gave their written, informed consent.

We registered our hypotheses and an analysis plan at the Open Science Framework: https://doi.org/10.17605/OSF.IO/H9SW6. As of the date of registration, both data sets existed and were accessible to the authors. However, the dependent variable had not been constructed, and no analyses had been conducted in relation to the hypotheses of this study. The authors' prior knowledge about the data are described in more detail in the registration. The R and Stata code used in the analysis are available at the OSF project page: https://osf.io/zbxe4/.

In our empirical analysis, we use a three-level structural equation model with a probit regression in the structural model. The three levels are the solicitations, the donors, and the collection sites. To estimate level-specific effects of variables on level 2 and level 3 (there are no level 1 predictors), we group-mean center the level 2 variables, and include the group-means for variables of interest as predictors on level 3 [50, 51].

We first estimate a model including the covariates likely to be confounders of the relationship between WOM recruitment (H1a) or talking about donations (H2a) and compliance with solicitations. On the individual level, these are experience as a blood donor, GST, blood type, working hours, age, sex, and having children. On the collection site level, these are the sociodemographic characteristics average age, proportion of males, average experience, and whether a collection site is mobile or fixed.

In the second model, we add variables that might be confounders, but that could alternatively be mediators (and in the latter case should not be included as covariates). On the individual level, we additionally include affective attitudes, awareness of need, wanting more or less invitations, and satisfaction with the blood bank. On the collection site level, we further include wanting less/more solicitations and satisfaction with the blood bank as covariates. Below, we focus on the results from Model 2 to account for potential additional confounders, but results for the variables of interest do not differ substantially between Models 1 and 2 (see S1 Table).

In the third model, we add the interaction term between talking about donations and altruistic values to test hypothesis 4. Because the possibility to estimate an interaction using latent

variables in a three-level model was not integrated in Mplus version 8.0, we use mean-scores for altruistic values in place of the latent variable to test this hypothesis.

In the fourth model, we extend model 2 to include the interaction term between talking about donations and the mobile collection site dummy to test hypothesis 3. We further include a random slope for talking about donations [52].

We use Bayesian inference with Markov Chain Monte Carlo (MCMC) estimation in Mplus version 8 [53] via the MplusAutomation package [54] for R [55]. MCMC estimation enables the estimation of a three-level structural equation model with a binary dependent variable and latent independent variables. Our estimation used 4 chains and the default priors used by Mplus [53]. Chains were run for a minimum of 20 000 iterations, and until convergence was achieved as indicated by all parameters having a potential scale reduction factor (PSRF) lower than 1.05 [56]. Convergence of the models was further assessed by inspecting trace plots and autocorrelation plots. The first half of iterations was discarded for burn-in.

We calculate average marginal effects (AMEs) for the variables of interest to be able to assess the strength of association on the probability scale. To do so, we re-estimate the models using unit-weighted mean-scores instead of the latent variables and maximum likelihood (ML) estimation in Stata version 16 [57]. We compare the probit coefficients between the two approaches to confirm that results are robust to different estimation procedures (see S1 and S2 Tables). The results are substantially the same, which makes us confident that neither the use of listwise deletion and mean-scores instead of latent variables (two limitations of the ML-estimation) or uncertainty about convergence (a limitation of the MCMC estimation) affect our results.

Below, the abbreviation 95% CI refers to *95% credible intervals* when referring to results based on Bayesian MCMC estimation, and to *95% confidence intervals*, when referring to results based on ML estimation.

## 5. Results

We hypothesized that compliance will be higher for donors that are recruited via WOM (H1a), and for donors that talk more about blood donations (H2a), and that compliance rates will be higher at collection sites with a higher proportion of donors recruited via WOM (H1b), and with a higher average level of talking about donations (H2b). Fig 4 shows the level of compliance differentiated by recruitment via word of mouth and by the level of talking about donations. On the individual level, there is almost no difference between the compliance rate of donors that were recruited via WOM and those that were recruited via other channels (see Fig 4A). In contrast, there is a clear positive bivariate association between compliance and the level of talking about donations (see Fig 4B). Donors that never talk about blood donations have about a 50% compliance, donors that occasionally talk about blood donations have about a 59% compliance, donors that regularly talk about donations have about a 67% compliance, and the very small group of donors that often talk about blood donations have about a 59% compliance with solicitations for donations.

A similar picture emerges on the collection site level. We do not see an association between the level of compliance and the proportion of donors recruited via WOM (Fig 4C), while there is a clear positive bivariate association between the compliance rate and the level of talking about donations (Fig 4D).

### 5.1. Results of statistical models

Results of the statistical models are shown in S1 Table (MCMC estimation) and S2 Table (ML estimation), and average marginal effects are depicted in Fig 5. As outlined in section 4, we

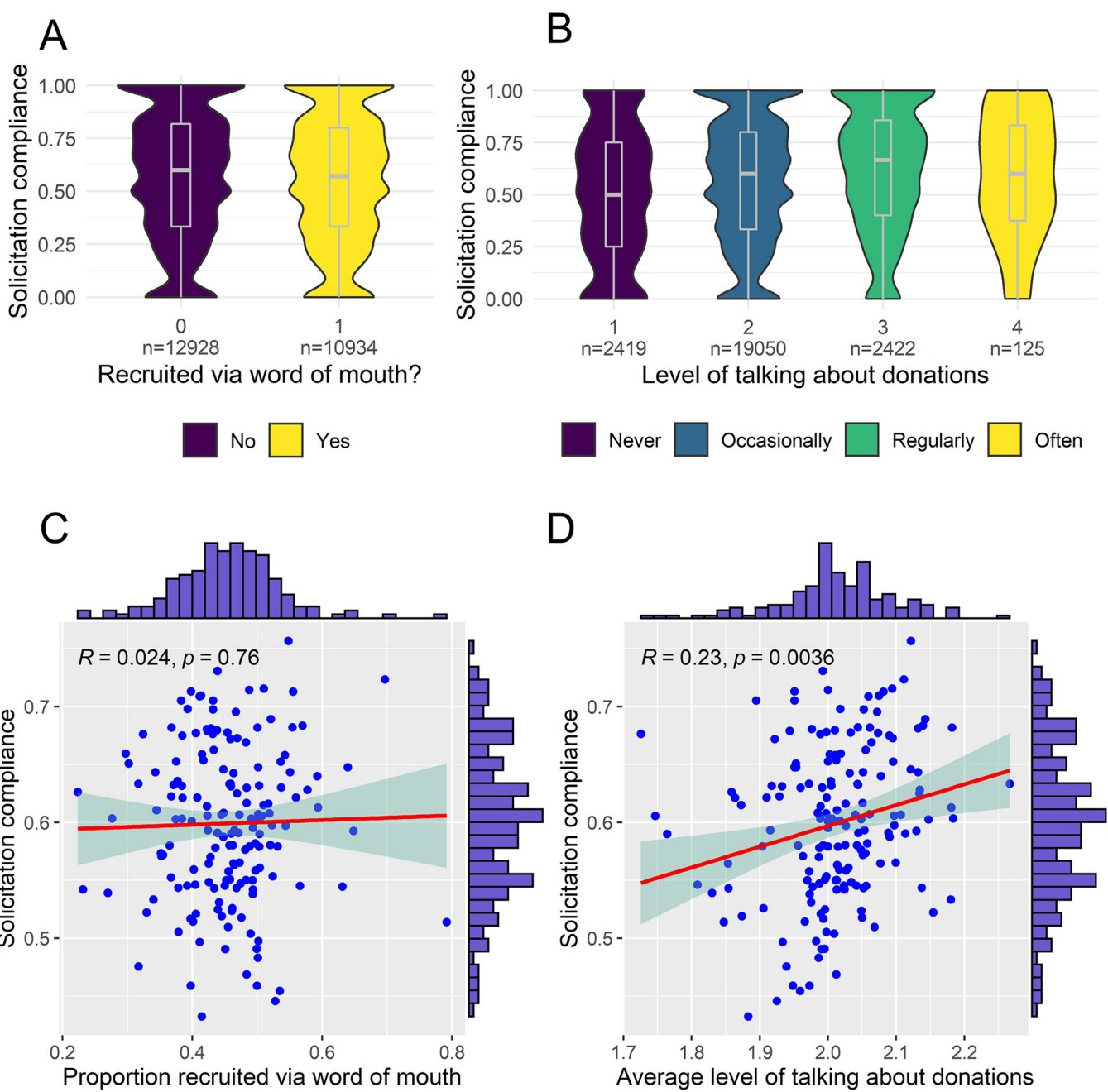

**Fig 4. Level of compliance by recruitment via word of mouth and by the level of talking about donations on the individual level and the collection-site level.** (A) Level of compliance for donors that were recruited via WOM or not, N = 23 862, missing = 183. (B) Level of compliance by level of talking about donations, N = 24 016, missing = 29. (C) Level of compliance by WOM recruitment on collection site level, N = 164. (D) Level of compliance by level of talking about donations on collection site level, N = 164.

focus on Model 2 for the test of hypotheses 1, 2, and 5, and Models 3 and 4 for the test of hypotheses 3 and 4.

## Perception of the social context

In line with the descriptive statistics, the model estimates that the association between WOM recruitment and compliance is essentially zero (b = -0.001, 95% CI = -0.023, 0.020), as also

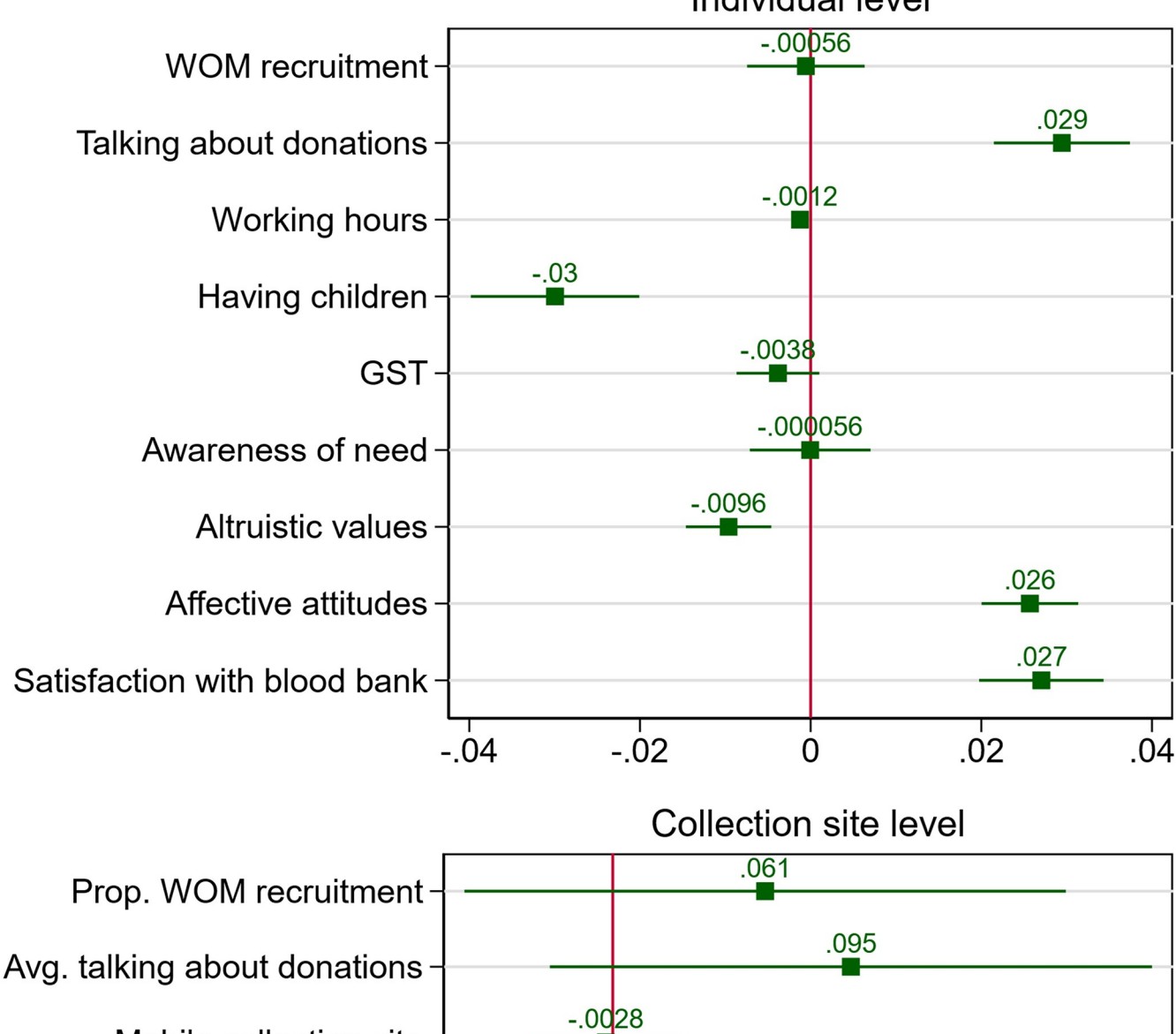

**Fig 5. Coefficient plot of average marginal effects (AMEs) on the probability of compliance with a solicitation for a donation.** AMEs are based on model 2 in S2 Table.

shown by the AME of zero (see Fig 5). Donors that were recruited via WOM are therefore not more likely to comply than donors that signed up on their own initiative or those that were recruited by the blood bank. Hypothesis 1a is thus not supported by the data.

Talking about donations is positively associated with compliance (b = 0.083, 95% CI = 0.061, 0.106). Donors that are one unit higher on the measure of talking about donations are estimated to have a 2.9 percentage points higher probability of compliance with a solicitation for a donation, net of other key determinants of compliance behaviour (see Fig 5). This is in line with hypothesis 2a.

On the collection site level, both WOM recruitment (b = 0.206, 95% CI = -0.174, 0.578) and talking about donations (b = 0.264, 95% CI = -0.077, 0.603) are most likely positively associated with compliance, but both positive and smaller negative associations are plausible. Hypotheses 1b and 2b are therefore not sufficiently supported by the data.

### Evaluation of the context

Model 4 in S1 Table shows the results for hypothesis 3, which stated that the individual-level association between talking about donations and compliance should be stronger at mobile than fixed collection sites. However, we do not find substantial variation in the association between talking about donations and compliance across collection sites (see S2 Fig); a likelihood-ratio test reveals that the inclusion of a random slope for talking about donations does not substantially increase the model fit ($\chi^2(1)$ = 0.17, p = 0.685). The posterior distribution of the estimate for the cross-level interaction provides some support for the positive interaction (b = 0.025, 95% CI = -0.029, 0.077), but also shows that both negative and positive interactions are plausible based on our data. Hypothesis 3 is therefore not supported by the data.

With Hypothesis 4 we hypothesised that the association between talking about donations and compliance might be stronger for those with lower altruistic values. The results of model 3 show the expected negative interaction (see S1 Fig), but the 95% credibility interval indicates that no interaction effect or a positive interaction effect are also plausible (b = -.018, 95% CI = -0.049, 0.012). Hypothesis 4 is therefore not supported by the data.

### Selection of behaviour

Hypothesis 5 stated that the association between talking about donations and compliance should be weaker for more experienced donors. This hypothesis is supported by the negative interaction effect between talking about donations and the number of previous donations (b = -.001, 95% CI = -0.002, -0.001). For the least experienced donors, a one-unit increase in talking about donations is predicted to increase the probability of compliance by about 5 percentage points (see Fig 6). For donors with more experience, in contrast, the average marginal effect of talking about donations is much smaller, and even turns negative for the very few donors with a very large number of previous donations.

### 5.2. Robustness checks and exploratory analyses

We conducted the following non-registered robustness checks and exploratory analyses to assess the reliability of our results and provide further insights into what might drive compliance with solicitations for donations.

### Robustness checks

A concern regarding our result for hypothesis 2a might be simultaneity; i.e., reciprocal causation between talking about donations and compliance. Our data does not allow us to cleanly disentangle these two aspects, and it is not the goal of this study to provide an estimate for the causal effect of talking about donations on compliance. However, we can provide two pieces of evidence that talking about donations has predictive power with regards to compliance. First, we re-estimate Model 2 using a restricted sample of donors that participated in the survey in 2012 and using compliance in 2013 as the dependent variable (see S4 Table Model 1). That is, we regress compliance in 2013 on talking about donations in 2012. The estimate from this approach is similar, albeit smaller than the one from our main analysis. A one-unit increase in talking about donations is associated with a 1.7 percentage point increase in the probability of

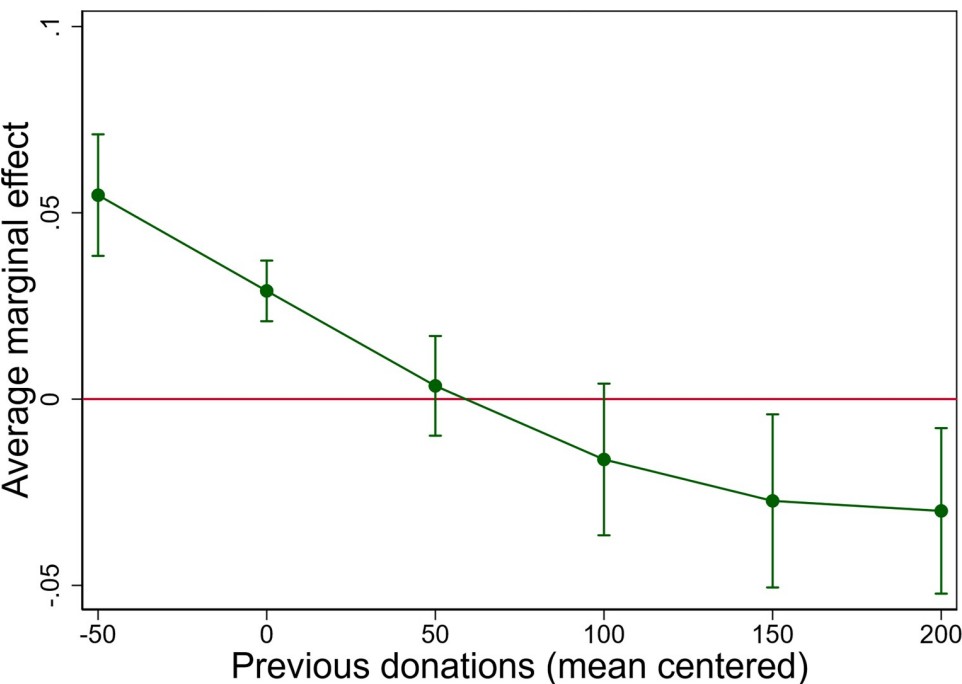

**Fig 6. Average marginal effect of talking about donations with 95% CI by level of experience as a blood donor.** AMEs are based on model 2 in S2 Table.

compliance (b = 0.051, 95% CI = 0.015, 0.087, AME = 0.017). This test does not provide a strict test of causality since it does not rule out that talking about donations is affected by prior levels of compliance to some extent. It does however show, that talking about donations can be used to predict compliance in the future. Second, to further assess the question of causality, we re-estimate Model 2 in a restricted sample of donors that did not make any donations prior to participating in the DIS-II survey (see S4 Table Model 2). For these donors, the extent to which they talk about donations should be exogenous with respect to compliance, since they do not yet have a level of compliance. This subsample is small, including only 642 donors, so statistical power is low. However, the estimate for the association between talking about donations and compliance in the following years is also positive and similar in size to the one using the lagged talking about donations specification, but the confidence interval does include zero (b = 0.056, 95% CI = -0.102, 0.214, AME = 0.019).

Next, taking account of recent recommendations to use linear models for the analysis of binary outcomes [58], we further show the robustness of our results to using a multilevel linear probability model (S5 Table) and OLS regression with standard errors clustered at the collection site level (S6 Table).

Finally, we show that our results for the association between WOM recruitment and compliance are robust to an alternative construction of the WOM recruitment variable described in section 3.2. (see S7 Table Model 1), and that the association between talking about donations and compliance is robust to the exclusion of those who 'often' talk about blood donations, which might be employees of the blood bank (see S7 Table Model 2).

## Results from exploratory analyses

Our models also reveal interesting results for factors that were identified as potentially important to compliance with solicitations for donations based on previous literature (see Fig 5).

First, potential costs of compliance, such as higher working hours (b = -0.004, 95% CI = -0.005, -0.003) and having children (b = -0.095, 95% CI = -0.122, -0.068), show an expected negative association with the probability of compliance. Next, general values often associated with prosocial behaviour do not seem to predict compliance well. Neither awareness of need (b = -0.003, 95% CI = -0.021, 0.015) nor GST (b = -0.017, 95% CI = -0.038, 0.002) are strongly associated with the probability of compliance. Altruistic values are even negatively associated with compliance (b = -0.058, 95% CI = -0.088, -0.028). Finally, blood donation specific attitudes show strong positive associations with compliance behaviour. A donor one unit higher on affective attitudes is predicted to have a 2.6 percentage points higher compliance (b = 0.087, 95% CI = 0.071, 0.104; AME = 0.026), and a donor that is one unit higher on satisfaction with the blood bank is predicted to have a 2.7 percentage points higher compliance with solicitations for donations (b = 0.105, 95% CI = 0.078, 0.133; AME = 0.027).

Finally, we assessed whether WOM recruitment, like talking about donations, might be more relevant for the compliance of novice donors. However, this interaction effect was not supported by the data (b = 0.000, 95% CI = -0.000, 0.001).

## 6. Discussion and conclusion

This study has analysed to what extent compliance with solicitations for blood donations is related to talking about donations and WOM recruitment at both the individual and the collection site level. Our results show that higher talking about donations indeed predicts higher compliance. This association was moderated by donor experience, such that it is strongest for novice donors with few previous donations. Conversely, being recruited via WOM did not predict higher compliance. In addition, our data did not support the hypotheses that talking about donations and WOM recruitment explain compliance rates on the collection site level.

We developed and tested a SES model of compliance, where a donor's decision-making process consists of the perception of their social and physical environment, the evaluation of new information, and the selection and execution of a behaviour. Based on our results, talking about donations emerged as an important component of a donors' perception of their social environment that is predictive of their compliance behaviour. This is in line with previous literature showing that contributions to public goods are higher in experimental settings where (potential) contributors can communicate [15, 16]. Our study extends this literature in two ways: First, our results suggest that it is the individual perception of the context via talking about donations that is pivotal for compliance rather than the broader social context. On the level of collection sites, which pose a relevant social and physical context for blood donations, differences in compliance rates do not seem to be explained by differences in WOM recruitment in talking about donations when the socio-demographic composition of the donor population is taken into account. Other social contexts, such as individuals' social networks, might instead be more relevant for compliance behaviour. Second, our study shows that the importance of communication translates to the case of compliance with solicitations for donations rather than general prosocial behaviour.

Regarding evaluation of the social context, our data did not support the hypothesis that the association between talking about donations and compliance is stronger at mobile collection sites, where donors are more likely to interact with others that are socially close to them. Talking about donations might therefore capture more general communication with other social network members rather than the immediate communication with others donors at the point of making a donation. Further, we do not find conclusive evidence that the evaluation depends on individuals' altruistic values. Based on our data, negative interaction effects but also small positive interaction effects are plausible. The crucial difference to previous studies of this

interaction [22, 23] is our focus on compliance rather than general prosocial behaviour: blood donors self-select into being a blood donor based on their altruistic values [59], and hence there is limited variation in donors' altruistic values, and altruistic values are overall negatively associated with compliance.

Regarding the process of selecting a behaviour, we find support for predictions derived from theories of habit formation in blood donor behaviour [36, 38, 39]: the interaction effect between talking about donations and experience as a blood donor indicates that talking about donations is particularly important for novice donors.

Being recruited via WOM, on the other hand, does not seem to be an element of donors' perception that influences their compliance behaviour. For many donors, the recruitment process may already be too far in the past to be relevant for their contemporary compliance behaviour. For example, they might no longer be in contact with the person(s) that motivated them to become a donor, and therefore no longer be subject to their social influence. Our data, however, does not indicate that WOM recruitment results in higher compliance of novice donors.

## 6.1. Implications

The primary contributions of our study to theory are to recognise solicitations for donations as a distinct level of analysis, and to demonstrate the social embedding of the donors' decision-making processes about compliance with such solicitations. Much of the prosocial behaviour we observe in the real world is the result of compliance with solicitations for donations rather than spontaneous giving. A shift in focus on compliance rather than general prosocial behaviour can therefore be useful to more accurately capture how decisions about contributions to public goods are made. The application of a SES model is a further step towards a better understanding of these decisions. For example, our SES model allows us to examine how talking about donations is an important feature linking the donors decision-making process to their social environment, while maintaining a comprehensive micro-level framework for understanding the human decision-making process. The importance of the distinction between compliance and prosocial behaviour in general is further highlighted by evident dissimilarities in factors associated with compliance versus general prosocial behaviour. Our analysis shows that values often seen as conductive to prosocial behaviour, namely awareness of need, generalized social trust and altruistic values, are not necessarily associated with an individual's compliance with solicitations for donations. Our speculative interpretation of these findings is that these factors do not play a large role for compliance because blood donors already self-select into becoming a donor based on these values.

A practical implication of our findings relates to the organizations that are dependent on the effectiveness of solicitations for donations, and blood banks in particular. Blood has a limited shelf life and demand is changing continuously. Knowing about the factors that determine compliance with solicitations is therefore essential to ensure a sufficient blood stock. Our results imply that increasing talking about blood donations among current donors could be one tool to increase the effectiveness of solicitations for donations. One strategy to achieve that may be group-donation programmes [60], where donors form groups that they can communicate and donate with–a promising area for future research. Such groups should create some actual feeling of relatedness, as previous studies have shown that social influence is at work among closely related individuals [17, 61, 62], but not among distant peers [63]. Finally, promoting communication about donations might be particularly effective for donors in early stages of their donor career, since their behaviour is more malleable than that of very experienced donors.

## 6.2. Limitations

The main limitation of our study is that its results cannot be interpreted causally. There are two main threats to a causal interpretation: First, it might be that individuals that talk more about donations have a higher compliance due to unobserved confounders. And second, it is likely that talking about donations is partly caused by compliance. For example, reputational concerns could mean that individuals are more likely to talk about donations when they generally comply with solicitations than when they do not. We have conducted two robustness checks which alleviate the implications of these concerns for practice, as they show that talking about donations is predictive of future compliance. Blood banks could either implement strategies that increase communication about blood donations to achieve higher compliance, or strategies that target recruitment at donors that are more likely to talk about donations in the first place.

Another limitation of our study is that we do not have insights into the potential mechanisms of social influence that might explain an effect of talking about donations on compliance. Previous research suggests that information about need, descriptive norms, and group identities might be underlying mechanisms [15, 16], but we cannot differentiate between these mechanisms in this study.

## 6.3. Directions for future research

An important direction for future research is to provide causal evidence on the effect of increased opportunities for communication on compliance behaviour. This includes research on its potentially reciprocal causation with compliance, the mechanisms that talking about donations might operate through, and the most effective ways to implement the findings of this research into practice via corresponding retention strategies.

Another important contribution of future research could be a complete analysis of the SES model suggested in this article. This article has focussed on how individual behaviour is shaped by the social context, but another important question in SES analysis is how the social and physical context emerges from the behavioural choices of individuals. This question is particularly relevant when considering processes of social influence in the long run, because there will be reciprocal causality between individual's decision and the social context, in this case the decisions and attitudes of other people.

## Supporting information

**S1 Table. Three-level structural equation model: regression of compliance on individual and collection site characteristics.**
(DOCX)

**S2 Table. Three-level probit regression of compliance on individual and collection site characteristics.**
(DOCX)

**S3 Table. Pearson correlation coefficients among study measures on the individual level.**
(DOCX)

**S4 Table. Robustness checks on the predictive power of talking about donations.**
(DOCX)

**S5 Table. Three-level linear probability regression of compliance on individual and collection site characteristics.**
(DOCX)

**S6 Table. OLS regression of compliance on individual and collection site characteristics.**
(DOCX)

**S7 Table. Robustness checks: Alternative construction of main predictors.**
(DOCX)

**S1 Fig. Average marginal effect of talking about donations with 95% CI by level of altruistic values.** AMEs are based on model 3 in S2 Table.
(DOCX)

**S2 Fig. Average marginal effect of talking about donations with 95% CI by type of collection site.** AMEs are based on model 4 in S2 Table.
(DOCX)

**S1 File.**
(TIF)

## Acknowledgments

We thank Mauricio Garnier-Villarreal for valuable remarks on an earlier version of the manuscript. We acknowledge valuable comments and suggestions from participants at the 2021 European Conference on Donor Health and Management (ECDHM), the 2021 Conference of the European Research Network on Philanthropy (ERNOP), and the 2021 Annual Conference of the Association of Research on Nonprofit Organizations and Voluntary Action (ARNOVA).

## Author Contributions

**Conceptualization:** Joris Melchior Schröder, Eva-Maria Merz, Bianca Suanet, Pamala Wiepking.

**Data curation:** Joris Melchior Schröder.

**Formal analysis:** Joris Melchior Schröder.

**Funding acquisition:** Eva-Maria Merz.

**Investigation:** Joris Melchior Schröder, Eva-Maria Merz, Bianca Suanet, Pamala Wiepking.

**Methodology:** Joris Melchior Schröder.

**Project administration:** Joris Melchior Schröder.

**Supervision:** Eva-Maria Merz, Bianca Suanet, Pamala Wiepking.

**Validation:** Joris Melchior Schröder.

**Visualization:** Joris Melchior Schröder.

**Writing – original draft:** Joris Melchior Schröder.

**Writing – review & editing:** Joris Melchior Schröder, Eva-Maria Merz, Bianca Suanet, Pamala Wiepking.

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
