## [Decision Letter · Decision Letter 0]

18 Oct 2022

PONE-D-22-20578Did you donate? Talking about donations predicts compliance with solicitations for donationsPLOS ONE

Dear Dr. Schröder,

Thank you for submitting your manuscript to PLOS ONE. After careful consideration, we feel that it has merit but does not fully meet PLOS ONE’s publication criteria as it currently stands. Therefore, we invite you to submit a revised version of the manuscript that addresses the points raised during the review process. The paper address an important question and is well written. However, there are two major concerns, regarding the two key variables of interest. As one of the reviewers discuss, the first WOM recruitment variable is problematically constructed. Moreover, there is a strong problem with the talking variable, which is clearly endogenous. I am not sure if you could fix this two important points in your revision, but I consider they must be address. There are some other comments by the reviewers. Please, read them carefully and try to follow their recommendations.

We look forward to receiving your revised manuscript.

Kind regards,

Alfonso Rosa Garcia

Academic Editor

PLOS ONE

Journal Requirements:

Reviewers' comments:

Reviewer's Responses to Questions

**Comments to the Author**

1. Is the manuscript technically sound, and do the data support the conclusions?

Reviewer #1: Yes

Reviewer #2: No

2. Has the statistical analysis been performed appropriately and rigorously? 

Reviewer #1: Yes

Reviewer #2: No

3. Have the authors made all data underlying the findings in their manuscript fully available?

Reviewer #1: No

Reviewer #2: Yes

4. Is the manuscript presented in an intelligible fashion and written in standard English?

Reviewer #1: Yes

Reviewer #2: Yes

5. Review Comments to the Author

Reviewer #1: What makes solicitations to make a donation for repeat donor effective? This paper conducts a set of registered analyses on Dutch data to show that some characteristics of donors, which can cheaply be measured by collection agencies, are drivers of compliance with solicitations. Testing a sociological theory with pre-registered hypotheses, the authors find that whether donors talk often about blood donations predicts compliance but the mode of recruitment (was the donor recruited via word-of-mouth?) does not.

I like this paper a lot: it addresses an important question for blood collection, but also repeated donations more generally; it gathers high quality administrative data on donation solicitation and actual donation and combines this to a large-scale survey effort; the manuscript is written clearly and elegantly. I recommend the paper to be published following some revisions.

1. For readers that are skeptical of structural assumptions, it would good if all the main analyses could be replicated using OLS. It’d be enough for each estimated model had an OLS counterpart.

2. The authors claim that the study was pre-registered. However, as they explain in their “pre-registration”, they already had access to these datasets prior to submitting the pre-registration on OSF. I urge the authors to call this a “registration” instead of pre-registration, and if they want to write in the paper that registration document was submitted prior to any analyses they should always be clear that at the time they however had access to the entire dataset already.

3. Often blood donors work in the health sector themselves or at the NGO of blood collection. This kind of folks would naturally talk more about blood donations, have easier access to donation centers, and feel more obliged when they receive the request -- which wouldn't be very surprising. Does the survey collect information on the job of the subjects? It would useful and important to determine whether the correlation between compliance and talking about blood is not entirely driven by this aspect. If this were the case, the implication that "Our results imply that increasing talking about blood donations among current donors could be one tool to increase the effectiveness of solicitations for donations." would not be warranted. So please either provide evidence that this headline correlation is not driven by the donor's job or at the very least drop the discussion of unwarranted implications.

Reviewer #2: This paper combines register and survey data to examines to what extent compliance with blood donation solicitations is predicted by being recruited via word of mouth and talking about donations. It is important to have a deeper understanding of what drives the compliance. However, I have some concerns about the two variables of interest. Please find my comments below.

Figure 1: There is a typo: Donors are in the Level 2 instead of level 3.

Variables of interest

First of all, I would not name Section 3.2 “independent” variables of interest, as the authors also admit in the limitation, these variables are not independent, for example, talking about donations could be the outcomes of donations.

Recruitment channel by WOM vs by the blood bank

Are these two channels mutually exclusive? Since you allow multiple options in the survey, it’s not completely clear how the WOM recruitment variable is constructed. If a survey participant chooses both 3) blood bank and 6) partner, would it count as WOM or not? And by “one of the options 6,7, or 8”, does it mean “exactly one” or “one and above”? Please provide unambiguous instructions of how this variable is constructed.

Talking about donations

By construction, if people are recruited by WOM, they have to talk about donations. This makes it extremely hard to rationalize the results that talking about donations matters while WOM recruitment doesn’t.

Why are the numbers of observation different in Figure 4A and 4B?

4A: N_0=12928, N_1=10934, total=23862

4B: N_1=2419, N_2=19050, N_3=2422, N_4=125, total=24016

Type of collection site: fixed vs mobile

Is it the type of residents in small towns and villages or is it the type of communication among people that makes the difference between the fixed and mobile? Please provide more underlying explanations for this hypothesis.

Discussion and limitation

The authors proposed an explanation that “the recruitment process may be too far in the past to be relevant for their contemporary compliance behaviour.” However, the same could apply to “talking about donations”, i.e. they may have talked about it a while ago. Yet, talking appears to matter while WOM recruitment doesn’t. It is hard to investigate it as both the WOM recruitment variable and talking variable lack a time reference with respect to the donations.

Regarding the process of selecting a behaviour, it’s not clear how you find support for habit formation simply by looking at the interaction between talking and experience. Please elaborate.

Why do people talk about donations in the first place? Maybe because these people are more likely to donate anyway, and therefore, what you observe is a pure correlation between motivated donor and the compliance of donations. Similarly, the proposed group donation programmes, it could attract donors who are motivated anyway, leading to the compliance of joining such group programmes which further influence the compliance of the final donation.

Finally, most of your hypotheses are not supported with your data, which calls for more appropriate measurement and methods.

6. PLOS authors have the option to publish the peer review history of their article (what does this mean?). If published, this will include your full peer review and any attached files.

Reviewer #1: No

Reviewer #2: No

---

## [Author Response · Author response to Decision Letter 0]

2 Dec 2022

Dear Prof. Rosa-Garcia,

Thank you very much for the opportunity to revise and resubmit our manuscript entitled "Did you donate? Talking about donations predicts compliance with solicitations for donations". We greatly appreciate the time you and the reviewers dedicated to reviewing our manuscript, and are thankful for the feedback and valuable comments we received.

Based on the reviews, we have conducted a revision of our manuscript. We provide a response to these comments below and in the attached file 'Response_to_Reviewers.docx'. The most important revisions concern the two points you mention in your decision letter:

1) We have clarified the construction of the variable measuring recruitment via word of mouth (WOM) in the paper, and conducted a robustness check using an alternative construction of the variable to address the concerns raised by Reviewer 2. We have detailed these changes and explained the additional analyses in our response to Reviewer 2’s comments below.

2) We have conducted two additional robustness checks to alleviate your concern about the endogeneity of talking about donations and compliance. In our view, the specific endogeneity issue is simultaneity: that talking about donations and compliance will, to some extent, cause one another, as previously noted in the study’s limitations. While it is not the goal of this study to provide an estimate for the causal effect of talking about donations on compliance with solicitations for donations, we agree that it is important to establish that talking about donations is not entirely caused by compliance itself. Below, we summarise two robustness checks that suggest that this is not the case. We have also included these two additional robustness checks in the revised manuscript.

First, we have re-estimated the main model for the test of hypothesis 2a (on the association between compliance and talking about donations) using a restricted sample of donors that participated in the survey in 2012, and their compliance in the year 2013 (see section 5.2. and table S4, Model 1). That is, we regress compliance in 2013 on talking about donations in 2012. Through the use of this ‘lagged’ measure of talking about donations, the extent of talking about donations is exogenous with respect to the current level of compliance. The estimate from this approach is in the same direction, albeit a little smaller than the one from our main analysis. A one-unit increase in talking about donations is associated with a 1.7 percentage point increase in the probability of compliance (b = 0.051, 95% CI = 0.015, 0.087, AME = 0.017).

This robustness check does not provide a strict test of causality, since it does not rule out that talking about donations is to some extent affected by prior levels of compliance. It does show, however, that talking about donations in one period can be used to predict compliance in the future. 

Second, we restricted our sample to those donors that did not make any donations prior to participating in the DIS-II survey (see table S4, Model 2). For these donors, the extent to which they talk about donations should be plausibly exogenous with respect to compliance, since they do not yet have a level of compliance. Unfortunately, this subsample is small, including only 642 donors. However, the estimate for the association between talking about donations and compliance in the following years is also positive and similar to the the one from the robustness check explained above, but the confidence interval does include zero (b = 0.056, 95% CI = -0.102, 0.214, AME = 0.019).

In our view, these robustness checks provide evidence that talking about donations has predictive power with regards to compliance with solicitations for donations. An in-depth analysis of the reciprocal causal relationship between talking about donations and compliance will be an important area for future studies, as we note in the discussion section of the study.

In order to give a detailed explanation of how we have dealt with the reviewers’ comments, we have included them below in the order we received them, and given our response to each comment.

Reviewer 1:

What makes solicitations to make a donation for repeat donor effective? This paper conducts a set of registered analyses on Dutch data to show that some characteristics of donors, which can cheaply be measured by collection agencies, are drivers of compliance with solicitations. Testing a sociological theory with pre-registered hypotheses, the authors find that whether donors talk often about blood donations predicts compliance but the mode of recruitment (was the donor recruited via word-of-mouth?) does not.

I like this paper a lot: it addresses an important question for blood collection, but also repeated donations more generally; it gathers high quality administrative data on donation solicitation and actual donation and combines this to a large-scale survey effort; the manuscript is written clearly and elegantly. I recommend the paper to be published following some revisions.

Thank you for your positive reflection on the merits of the article and the following comments, which have significantly contributed to improving our manuscript.

1. For readers that are skeptical of structural assumptions, it would good if all the main analyses could be replicated using OLS. It’d be enough for each estimated model had an OLS counterpart.

We appreciate the concern that some readers might be sceptical of certain assumptions underlying the statistical analyses. We interpret your raised issue about ‘structural assumptions’ to either refer to a) functional form assumptions underlying the non-linear probit models, or b) assumptions necessary for the inclusion of random effects in the multilevel models. We still consider the choice of a nonlinear multilevel structural equation model most suited, because it captures the specifics of the data, including a binary outcome, nested data structure, and latent variables. However, the following robustness checks can be used to validate the results under a different set of assumptions.

First, we re-estimate our models as multilevel linear probability models (ML-LPM) instead of multilevel probit models. Second, we estimate a series of linear probability models via OLS, and account for the nested structure of the data by clustering the standard errors at the level of collection sites. Since both of these models are linear, they might result in predicted probabilities below 0 or above 1, which of course are nonsensical. However, since the average compliance in our sample is 0.57, we do not expect too many predicted probabilities for compliance to fall outside the unit interval. This is confirmed after estimation, which shows that the average of predicted probabilities from the ML-LPM is 0.570 (min. = 0.163, max. = 1.139, std. dev. = 0.102), and the average of predicted probabilities from the LPM using OLS estimation is 0.568 (min. = 0.141, max. = 1.103, std. dev. = 0.107), with about only 21 (ML-LPM) and 29 (OLS) predicted probabilities being larger than 1. The bias and inconsistency in estimates from the linear models should therefore be low (Horrace and Oaxaca 2006), and they can be valid robustness-checks against the probit model. 

The results from both approaches confirm all results reported in the paper, with small deviations in the estimated strength of associations. Below, we briefly report those for our main variables of interest based on re-estimating our preferred specification in the paper (model 2). Regarding WOM recruitment, both models reveal no discernable association on the individual level (bML-LMP = -0.000, 95% CI = -0.007, 0.007; bOLS = -0.000, 95% CI = -0.007, 0.006), and no clear evidence for a positive relationship on the collection site level (bML-LMP = 0.064, 95% CI = -0.054, 0.183; bOLS = 0.116, 95% CI = -0.026, 0.258). For talking about donations, both models show a clear positive association on the individual level (bML-LMP = 0.029, 95% CI = 0.021 , 0.037; bOLS = 0.026, 95% CI = 0.018, 0.035), and, like in our main analysis, there is no clear evidence for a positive relationship on the collection site level (bML-LMP = 0.086, 95% CI = -0.033, 0.206; bOLS = 0.129, 95% CI = -0.023, 0.281).

We included a short description of these robustness checks (see section 5.2, page 21) and the estimation results as tables S4 and S5 into the paper.

2. The authors claim that the study was pre-registered. However, as they explain in their “pre-registration”, they already had access to these datasets prior to submitting the pre-registration on OSF. I urge the authors to call this a “registration” instead of pre-registration, and if they want to write in the paper that registration document was submitted prior to any analyses they should always be clear that at the time they however had access to the entire dataset already.

We agree that it is very important to be clear and open about prior knowledge about data and data access when conducting a study based on secondary data. This is one of the reasons why we value the preregistration of studies even for secondary data analysis. To increase transparency about the research process in the paper itself, we added the following sentence to the methods section: ‘As of the date of preregistration, both data sets existed and were accessible to the authors. However, the dependent variable had not been constructed, and no analyses had been conducted in relation to the hypothesis of this study. The authors' prior knowledge about the data are described in more detail in the preregistration. 

We are not aware of a distinction between ‘registration’ and ‘preregistration’, and do not feel like a change in terminology would increase transparency about the research process. We therefore stick to the term ‘preregistration’, which is more commonly used and also applicable for studies using secondary data (Nosek et al. 2018; Akker et al. 2021).

3. Often blood donors work in the health sector themselves or at the NGO of blood collection. This kind of folks would naturally talk more about blood donations, have easier access to donation centers, and feel more obliged when they receive the request -- which wouldn't be very surprising. Does the survey collect information on the job of the subjects? It would useful and important to determine whether the correlation between compliance and talking about blood is not entirely driven by this aspect. If this were the case, the implication that "Our results imply that increasing talking about blood donations among current donors could be one tool to increase the effectiveness of solicitations for donations." would not be warranted. So please either provide evidence that this headline correlation is not driven by the donor's job or at the very least drop the discussion of unwarranted implications. 

It is a good point that people working at blood banks or the health sector in general are likely overrepresented in the donor population, and might be particularly active and vocal donors. Unfortunately, the survey did not include information on the participants’ occupation. At the same time, we think that it is extremely unlikely that association between talking about donations and compliance is entirely driven by employees of Sanquin. In 2013, Sanquin had 2880 employees and 380 289 donors (Sanquin 2013). Assuming that all employees of Sanquin also donate blood, they would make up 0.7% of the donor population. In our sample of 24 045 donors, there might therefore be around 168 (24045*0.007) donors that are employed at Sanquin.

Nevertheless, even a small number of outliers might affect the results of statistical models. To further rule out this explanation, we therefore re-estimated our preferred model (model 2 in the manuscript) after excluding the 125 people who indicated that they ‘often’ talk about blood donations, which are most likely to be employees of Sanquin. The results do not change significantly, with a one-unit increase in talking about donations being associated with a 3.2 percentage point increase in compliance (b = 0.094, 95% CI = 0.068, 0.119, AME = 0.032).

In our view, the link between occupation and donorship is therefore not a major concern in the context of this study, but it is an interesting point for future research which could try to link survey data to administrative occupational data.

We included a short description of this robustness check (see section 5.2., page 21) and the estimation results in table S6 of the paper.

Reviewer 2: 

This paper combines register and survey data to examines to what extent compliance with blood donation solicitations is predicted by being recruited via word of mouth and talking about donations. It is important to have a deeper understanding of what drives the compliance. However, I have some concerns about the two variables of interest. Please find my comments below.

Thank you for your reflection on the relevance of the research question and the following comments, which have significantly contributed to improving our manuscript.

1. Figure 1: There is a typo: Donors are in the Level 2 instead of level 3.

Thank you for pointing out this error, the figure has been corrected.

2. Variables of interest

First of all, I would not name Section 3.2 “independent” variables of interest, as the authors also admit in the limitation, these variables are not independent, for example, talking about donations could be the outcomes of donations.

As you point out, we are in agreement that these variables cannot necessarily be considered exogenous. We chose the term ‘independent variables’, because it clearly conveys their function as predictors in the statistical models. However, we agree with your remark and changed the heading to ‘Predictors of interest’, which should convey both that these variables are rhs variables but not necessarily exogenous.

3. Recruitment channel by WOM vs by the blood bank

Are these two channels mutually exclusive? Since you allow multiple options in the survey, it’s not completely clear how the WOM recruitment variable is constructed. If a survey participant chooses both 3) blood bank and 6) partner, would it count as WOM or not? And by “one of the options 6,7, or 8”, does it mean “exactly one” or “one and above”? Please provide unambiguous instructions of how this variable is constructed. 

Thank you for pointing out the ambiguity in the description of how the variable was constructed. We updated the description of the variable to read: “WOM recruitment takes the value 1 if the respondent selected at least one [emphasis added] of the options 6, 7, or 8, and the value 0 otherwise.” (p. 12). 

Since multiple response options could be selected in the survey, the recruitment channels are not mutually exclusive. In practice, most of the people indicating that they were recruited via WOM do not select another recruitment channel. Out of 10934 donors indicating that they were recruited via mouth (potentially among other channels), 8514 donors (78%) indicated that they were recruited only via WOM. As a robustness check, we re-estimated our preferred model (model 2 in the manuscript) using a variable indicating whether a donor was exclusively recruited via WOM. The results are substantially the same as those from the analysis with the original construction of the variable, with no strong evidence for an association between WOM recruitment and compliance on either the individual level (b = -0.022, 95% CI = -0.046, 0.001) or the collection site level (b = 0.185, 95% CI = -0.184, 0.554). A description of this robustness check has been added to section 3.2, and the results are reported in table S7.

4. Talking about donations

By construction, if people are recruited by WOM, they have to talk about donations. This makes it extremely hard to rationalize the results that talking about donations matters while WOM recruitment doesn’t.

In the original version of the manuscript we have not been clear enough about the process of becoming a donor in the Netherlands, which led to some ambiguity in understanding the relationship between WOM recruitment and talking about donations. In the revised manuscript, we discuss why these predictors are not necessarily strongly related (see p. 12).

It is true that donors that were recruited via WOM will, at some point in time, also have talked about blood donations. However, having been recruited via WOM does not say much about the current level of talking about donations. This is also shown by a weak correlation between these two variables in the data, with a pearson correlation coefficient of only r = 0.02 (see table S3). We think that there are two main reasons for this. First, many donors have been donating with the blood bank for several years or even decades, as we clarify in the adapted description of the blood donor registration process (see below). As we also mention in the paper, the point of recruitment might therefore lie far in the past, while the survey question measuring talking about donations refers to the time the survey was administered (also see our further response to point 7 below). Second, while being recruited via WOM requires some talking about donations, it does not say anything about the extent of talking about donations. That is, whether donors occasionally, regularly, or often talk about blood donations.

We extended the description of the blood donor registration process at the beginning of section 2 to clarify that the recruitment channel refers to the initial decision about becoming a donor. It now reads: “Prospective donors first register to become a donor with the blood bank, for example after being recruited by a friend. After registration, donors undergo an initial health screening and, if they are eligible, are added to the donor database. Subsequently, donors are repeatedly solicited to make a donation at a specific collection site” (p. 4). 

In addition, we also added a brief discussion on why these predictors are not necessarily strongly related (see p. 12). 

5. Why are the numbers of observation different in Figure 4A and 4B?

4A: N_0=12928, N_1=10934, total=23862

4B: N_1=2419, N_2=19050, N_3=2422, N_4=125, total=24016

The differences in the number of observations in Figure 4A and 4B stem from the different number of missing observations for each of these variables, which are displayed in Table 1. To make this more transparent, we added the number of missing observations to the figure notes.

6. Type of collection site: fixed vs mobile

Is it the type of residents in small towns and villages or is it the type of communication among people that makes the difference between the fixed and mobile? Please provide more underlying explanations for this hypothesis.

In our view, it is the type of relationships that matter most. Social proximity has been shown to be a moderator of social influences, such as those potentially operating through communication with other donors (Goette and Tripodi 2022; Bond et al. 2012). We use the type of collection site as a proxy for social proximity to the others which donors might be talking to. In the empirical setting, what matters most is therefore that mobile collection sites are mainly used in less densely populated areas and that all donors in the area will be invited to donate on the same day, which means that talking about donations in these areas will take place among people who are more socially close. 

We extended and clarified the description of this hypothesis. We also spelled out the previously implicit assumption that social networks are spatially clustered. The description now reads: 

“Social proximity has been shown to moderate the effect of social influences [34], including for the case of blood donations [18]. While we have no specific information on who interacts with whom, we can use information about collection sites as a proxy for social closeness among (potentially) interacting donors. In the Netherlands, blood is collected at fixed and mobile sites. Fixed collection sites are placed in larger cities and have extended opening hours. Donors registered at fixed sites are invited to donate during a two-week walk-in period starting shortly after receiving the solicitation letter. Mobile collection sites are used to collect blood in less densely populated areas such as smaller towns and villages, and therefore draw on a smaller pool of donors than fixed collection sites. Because social networks are spatially clustered [35], these donors are more likely to know other donors donating at the same collection site. In addition, donors at mobile collection sites are invited to donate at a specific date rather than within a two-week walk-in period. Together, these factors imply that donors invited to a mobile site are more likely to meet and talk to other donors that they know. The relation between talking about donations and compliance might therefore be stronger at mobile rather than fixed donation sites.” (p. 8)

7. Discussion and limitation

The authors proposed an explanation that “the recruitment process may be too far in the past to be relevant for their contemporary compliance behaviour.” However, the same could apply to “talking about donations”, i.e. they may have talked about it a while ago. Yet, talking appears to matter while WOM recruitment doesn’t. It is hard to investigate it as both the WOM recruitment variable and talking variable lack a time reference with respect to the donations.

In relation to our response to point 4, we feel that the paper should have been more clear about the process of becoming a blood donor in the Netherlands, which we have amended as described under point 4. Together with the wording of the question “What made [emphasis added] you decide to become a donor?”, it should become clear that WOM recruitment refers to the initial recruitment into becoming a blood donor. In contrast, the question for ‘Talking about donations’: “How often do you speak with people in your circle of acquaintances about blood donation?” refers to the time the survey was administered and was elicited in the years that donors’ compliance was measured.

8. Regarding the process of selecting a behaviour, it’s not clear how you find support for habit formation simply by looking at the interaction between talking and experience. Please elaborate.

The paper does not intend to provide a general test of habit formation in blood donation behaviour, which it indeed cannot do. Rather, it tests the specific prediction that an external influence on behaviour in the form of talking about donations will become less important as behaviour becomes increasingly habitualised. 

We depart from an established result from the blood donation literature that blood donation behaviour becomes increasingly habitual as it is performed more often (Masser et al. 2008; Charng, Piliavin, and Callero 1988; Ferguson et al. 2012). Experience as measured by the number of previous donations is therefore commonly used as an indicator for habit formation. With increasing habit, the decision to donate blood might become so routine that it is made with little conscious deliberation. At this stage, external factors such as motivations derived from talking about donations likely play a minor role for the decision to donate or not, which is the specific hypothesis we test.

This nuance might not have been completely clear in the manuscript, and we have therefore clarified the description of this hypothesis in section 2.2. We have also slightly modified our summary of this finding in the discussion: “[...] we find support for predictions derived from [emphasis added] theories of habit formation in blood donor behaviour [35,37,38]: the interaction effect between talking about donations and experience as a blood donor indicates that talking about donations is particularly important for novice donors.” (p. 25).

9. Why do people talk about donations in the first place? Maybe because these people are more likely to donate anyway, and therefore, what you observe is a pure correlation between motivated donor and the compliance of donations. Similarly, the proposed group donation programmes, it could attract donors who are motivated anyway, leading to the compliance of joining such group programmes which further influence the compliance of the final donation.

We do not think that talking about donations is a generic measure of being a motivated blood donor. In our models, we include many sociodemographic and attitudinal variables that are likely to be confounders because they are known to influence compliance and might also affect the extent to which donors talk about donations. We are therefore confident that talking about donations captures something more than just being a motivated donor, or at least a specific component of being a motivated donor.

The question why donors talk about donations in the first place is an interesting one. Previous research suggests that communication about blood donation might be a reputational signal (Lyle, Smith, and Sullivan 2009). Research from the Netherlands further shows that donors can be encouraged to talk about donations with others, and that their willingness to do so depends on their own experiences with donations and the blood bank, their confidence in the ability to inform others about blood donations, and the anticipated reaction of the dialogue partner (Lemmens et al. 2010; 2008). In our data, we can also see that donors who know other donors also talk more about donations, as indicated by a positive correlation between knowing other donors and talking about donations (r=0.14, p<0.000). As noted in our discussion, we consider this question an important direction for future research, which should help put our findings into practice through recruitment and retention strategies.

It is certainly correct that a group-donation programme might primarily attract donors that are more motivated to donate in the first place. However, we do not consider this a concern to the potential effectiveness of such programmes. In fact, research on group-membership and prosocial behaviour has shown that groups of motivated donors (those that contribute anyway) are particularly successful in maintaining cooperation over time (Guido, Robbett, and Romaniuc 2019; Gächter and Thöni 2005). In these groups, it is particularly the reciprocal motivation of motivated donors which keeps them giving in the long run.

To clarify that these questions are not yet known, we added “– a promising area for future research” (p. 26) to the paragraph mentioning the group-donation programme in the implications section. 

10. Finally, most of your hypotheses are not supported with your data, which calls for more appropriate measurement and methods.

It is true that we are not able to confirm all of our hypotheses. However, we do not share the conclusion, that the failure to confirm hypotheses implies poor measurement and methods. To the contrary, the replication crisis across social sciences has shown that high hypotheses-confirmation rates are often the results of bad scientific practices, such as post-diction and p-hacking (Ioannidis 2005; OPEN SCIENCE COLLABORATION 2015). 

The measures in this study are tightly linked to the hypotheses, are elicited in a large survey, and tested in a data set with almost complete coverage of blood collection sites in the Netherlands, a large sample of donors, and complete data on their compliance behaviour over two years. In addition, our hypotheses, measures, and methods were preregistered. In our view, they provide a good test of the specific hypotheses in this study. 

If anything, we believe that the failure to confirm several hypotheses stems from a low base-rate (the a priori probability that our hypotheses are true, see Miller & Ulrich (2019)), which might be the results of little previous research on compliance with solicitations for (blood) donations.

With our response to your previous comments, we have tried to address several of your more specific concerns regarding measurement and methods, and implemented several changes in the paper to address them. We hope that these changes alleviate your general concerns about the quality of measurement and methods in this paper.

We hope that this letter addresses all comments and suggestions made by the reviewers. We again would like to express our gratitude for the time and effort dedicated to improving our manuscript.

Sincerely,

Joris M. Schröder

On behalf of all authors.

References:

Akker, Olmo R. van den, Sara Weston, Lorne Campbell, Bill Chopik, Rodica Damian, Pamela Davis-Kean, Andrew Hall, et al. 2021. ‘Preregistration of Secondary Data Analysis: A Template and Tutorial’. Meta-Psychology 5 (November). https://doi.org/10.15626/MP.2020.2625.

Bond, Robert M., Christopher J. Fariss, Jason J. Jones, Adam D. I. Kramer, Cameron Marlow, Jaime E. Settle, and James H. Fowler. 2012. ‘A 61-Million-Person Experiment in Social Influence and Political Mobilization’. Nature 489 (7415): 295–98. https://doi.org/10.1038/nature11421.

Charng, Hong-Wen, Jane Allyn Piliavin, and Peter L. Callero. 1988. ‘Role Identity and Reasoned Action in the Prediction of Repeated Behavior’. Social Psychology Quarterly 51 (4): 303–17. https://doi.org/10.2307/2786758.

Ferguson, Eamonn, Femke Atsma, Wim de Kort, and Ingrid Veldhuizen. 2012. ‘Exploring the Pattern of Blood Donor Beliefs in First-Time, Novice, and Experienced Donors: Differentiating Reluctant Altruism, Pure Altruism, Impure Altruism, and Warm Glow’. Transfusion 52 (2): 343–55. https://doi.org/10.1111/j.1537-2995.2011.03279.x.

Gächter, Simon, and Christian Thöni. 2005. ‘Social Learning and Voluntary Cooperation Among Like-Minded People’. Journal of the European Economic Association 3 (2/3): 303–14.

Goette, Lorenz, and Egon Tripodi. 2022. ‘Social Recognition: Experimental Evidence from Blood Donors’. SSRN Scholarly Paper ID 3886951. Rochester, NY: Social Science Research Network. https://doi.org/10.2139/ssrn.3886951.

Guido, Andrea, Andrea Robbett, and Rustam Romaniuc. 2019. ‘Group Formation and Cooperation in Social Dilemmas: A Survey and Meta-Analytic Evidence’. Journal of Economic Behavior & Organization 159 (March): 192–209. https://doi.org/10.1016/j.jebo.2019.02.009.

Horrace, William C., and Ronald L. Oaxaca. 2006. ‘Results on the Bias and Inconsistency of Ordinary Least Squares for the Linear Probability Model’. Economics Letters 90 (3): 321–27. https://doi.org/10.1016/j.econlet.2005.08.024.

Ioannidis, John PA. 2005. ‘Why Most Published Research Findings Are False’. PLoS Medicine 2 (8): e124.

Lemmens, Karin P. H., Charles Abraham, Robert a. C. Ruiter, Ingrid J. T. Veldhuizen, A. E. R. Bos, and Herman P. Schaalma. 2008. ‘Identifying Blood Donors Willing to Help with Recruitment’. Vox Sanguinis 95 (3): 211–17. https://doi.org/10.1111/j.1423-0410.2008.01079.x.

Lemmens, Karin P. H., Robert A. C. Ruiter, Charles Abraham, Ingrid J. T. Veldhuizen, and Herman P. Schaalma. 2010. ‘Motivating Blood Donors to Recruit New Donors: Experimental Evaluation of an Evidence-Based Behavior Change Intervention’. Health Psychology 29 (6): 601–9. https://doi.org/10.1037/a0021386.

Lyle, H. F., E. A. Smith, and R. J. Sullivan. 2009. ‘Blood Donations as Costly Signals of Donor Quality’. Journal of Evolutionary Psychology 7 (4): 263–86. https://doi.org/10.1556/jep.7.2009.4.1.

Masser, Barbara M., Katherine M. White, Melissa K. Hyde, and Deborah J. Terry. 2008. ‘The Psychology of Blood Donation: Current Research and Future Directions’. Transfusion Medicine Reviews 22 (3): 215–33. https://doi.org/10.1016/j.tmrv.2008.02.005.

Miller, Jeff, and Rolf Ulrich. 2019. ‘The Quest for an Optimal Alpha’. PLOS ONE 14 (1): e0208631. https://doi.org/10.1371/journal.pone.0208631.

Nosek, Brian A., Charles R. Ebersole, Alexander C. DeHaven, and David T. Mellor. 2018. ‘The Preregistration Revolution’. Proceedings of the National Academy of Sciences 115 (11): 2600–2606. https://doi.org/10.1073/pnas.1708274114.

OPEN SCIENCE COLLABORATION. 2015. ‘Estimating the Reproducibility of Psychological Science’. Science 349 (6251): aac4716. https://doi.org/10.1126/science.aac4716.

Sanquin. 2013. ‘Sanqin Annual Report 2013’. Amsterdam: Sanquin Blood Supply. https://www.sanquin.nl/binaries/content/assets/sanquinen/about-sanquin/annual-reports/sanquin-annual-report-2013.pdf.

---

## [Decision Letter · Decision Letter 1]

10 Jan 2023

PONE-D-22-20578R1Did you donate? Talking about donations predicts compliance with solicitations for donationsPLOS ONE

Dear Dr. Schröder,

Thank you for submitting your manuscript to PLOS ONE. After careful consideration, we feel that it has merit but does not fully meet PLOS ONE’s publication criteria as it currently stands. Therefore, we invite you to submit a revised version of the manuscript that addresses the points raised during the review process. Please submit your revised manuscript by Feb 24 2023 11:59PM. If you will need more time than this to complete your revisions, please reply to this message or contact the journal office at plosone@plos.org. Please include the following items when submitting your revised manuscript:A rebuttal letter that responds to each point raised by the academic editor and reviewer(s). You should upload this letter as a separate file labeled 'Response to Reviewers'.A marked-up copy of your manuscript that highlights changes made to the original version. You should upload this as a separate file labeled 'Revised Manuscript with Track Changes'.An unmarked version of your revised paper without tracked changes. You should upload this as a separate file labeled 'Manuscript'.If applicable, we recommend that you deposit your laboratory protocols in protocols.io to enhance the reproducibility of your results. Protocols.io assigns your protocol its own identifier (DOI) so that it can be cited independently in the future. For instructions see: https://journals.plos.org/plosone/s/submission-guidelines#loc-laboratory-protocols. Additionally, PLOS ONE offers an option for publishing peer-reviewed Lab Protocol articles, which describe protocols hosted on protocols.io. Read more information on sharing protocols at https://plos.org/protocols?utm_medium=editorial-email&utm_source=authorletters&utm_campaign=protocols.

We look forward to receiving your revised manuscript.

Kind regards,

Alfonso Rosa Garcia

Academic Editor

PLOS ONE

Journal Requirements:

Additional Editor Comments:

As a suggestion from a referee, it is important to note that this paper should be referred to as a registration. Please ensure that this point is included in the final version.

Reviewers' comments:

Reviewer's Responses to Questions

**Comments to the Author**

1. If the authors have adequately addressed your comments raised in a previous round of review and you feel that this manuscript is now acceptable for publication, you may indicate that here to bypass the “Comments to the Author” section, enter your conflict of interest statement in the “Confidential to Editor” section, and submit your "Accept" recommendation.

Reviewer #1: (No Response)

2. Is the manuscript technically sound, and do the data support the conclusions?

Reviewer #1: Yes

3. Has the statistical analysis been performed appropriately and rigorously? 

Reviewer #1: Yes

4. Have the authors made all data underlying the findings in their manuscript fully available?

Reviewer #1: No

5. Is the manuscript presented in an intelligible fashion and written in standard English?

Reviewer #1: Yes

6. Review Comments to the Author

Reviewer #1: I want to thank the authors for taking all comments seriously and adequately addressing them in the revised manuscript.

I only want to push back on the response to my comment on pre-registration. The distinction between "registration" and "pre-registration" should be a well known one at this point, at least since some journals (like the American Economic Associations') introduced policies that require some form of registration. See here an example of a paper that was only "registered": https://gautam-rao.com/pdf/HMRS.pdf

For a paper to be considered pre-registered, it's necessary that the authors do not have access to the data prior to the registration. It's not enough for the authors to claim that they had it but didn't use it or that they had it but hadn't yet constructed the exact outcome that they were going to use. Imagine the consequences it that was the standard for everybody...

I urge the authors and the editor to call this a registration. It's important that as a profession we don't pool together different things and dilute the meaning of labels that should signal research replicability.

7. PLOS authors have the option to publish the peer review history of their article (what does this mean?). If published, this will include your full peer review and any attached files.

Reviewer #1: No

---

## [Author Response · Author response to Decision Letter 1]

12 Jan 2023

Dear Prof. Rosa-Garcia,

Thank you very much for the opportunity to revise and resubmit our manuscript entitled "Did you donate? Talking about donations predicts compliance with solicitations for donations".

Based on the comments by reviewer #1, we now refer to the paper as registered rather than preregistered.

We again would like to express our gratitude for the time and effort dedicated to improving our manuscript.

Sincerely,

Joris M. Schröder

On behalf of all authors.

---

## [Editor Report · Decision Letter 2]

18 Jan 2023

Did you donate? Talking about donations predicts compliance with solicitations for donations

PONE-D-22-20578R2

Dear Dr. Schröder,

We’re pleased to inform you that your manuscript has been judged scientifically suitable for publication and will be formally accepted for publication once it meets all outstanding technical requirements.

Kind regards,

Alfonso Rosa Garcia

Academic Editor

PLOS ONE
---

## [Editor Report · Acceptance letter]

20 Jan 2023

PONE-D-22-20578R2 

Did you donate? Talking about donations predicts compliance with solicitations for donations 

Dear Dr. Schröder:

I'm pleased to inform you that your manuscript has been deemed suitable for publication in PLOS ONE. Congratulations! Your manuscript is now with our production department. 

Kind regards, 

on behalf of

Dr. Alfonso Rosa Garcia 

Academic Editor

PLOS ONE